# Measurement uncertainties in quantifying aeolian mass flux: evidence from wind tunnel and field site data

Ate Poortinga[1], Joep G.S. Keijsers[1], Jerry Maroulis[1,2] and Saskia M. Visser[3]

[1] Soil Physics and Land Management Group, Wageningen University and Research Center, Wageningen, The Netherlands
[2] Faculty of Science, Health, Education and Engineering, University of the Sunshine Coast, Maroochydore DC, Queensland, Australia
[3] Team Soil Physics and Land Use, Alterra, Wageningen University and Research Center, Wageningen, The Netherlands

## ABSTRACT

Aeolian sediment traps are widely used to estimate the total volume of wind-driven sediment transport, but also to study the vertical mass distribution of a saltating sand cloud. The reliability of sediment flux estimations from such measurements are dependent upon the specific configuration of the measurement compartments and the analysis approach used. In this study, we analyse the uncertainty of these measurements by investigating the vertical cumulative distribution and relative sediment flux derived from both wind tunnel and field studies. Vertical flux data was examined using existing data in combination with a newly acquired dataset; comprising meteorological data and sediment fluxes from six different events, using three customized catchers at Ameland beaches in northern Netherlands. Fast-temporal data collected in a wind tunnel shows that the median transport height has a scattered pattern between impact and fluid threshold, that increases linearly with shear velocities above the fluid threshold. For finer sediment, a larger proportion was transported closer to the surface compared to coarser sediment fractions. It was also shown that errors originating from the distribution of sampling compartments, specifically the location of the lowest sediment trap relative to the surface, can be identified using the relative sediment flux. In the field, surface conditions such as surface moisture, surface crusts or frozen surfaces have a more pronounced but localized effect than shear velocity. Uncertainty in aeolian mass flux estimates can be reduced by placing multiple compartments in closer proximity to the surface.

## INTRODUCTION

Aeolian sediment transport is an important geomorphological process that shapes a number of landscapes including coastal (e.g., *Arens, 1996*; *Wal, 2000*; *Jackson & Nordstrom, 2011*), drift sand (e.g., *Riksen et al., 2006*; *Riksen & Goossens, 2007*), deserts (e.g., *Bagnold, 1941*; *Wiggs, 2001*), and also agricultural areas (e.g., *Visser & Sterk, 2007*; *Chepil & Woodruff, 1963*; *Visser & Sterk, 2007*). Along sandy coasts, aeolian processes drive the

Corresponding author
Ate Poortinga,
ate.poortinga@wur.nl

morphological development of coastal dunes that protects the hinterland against flooding. Maintaining the natural aeolian dynamics allows vegetation to flourish in different successive stages, creating an appealing area for tourism and recreation (*Poortinga et al., 2011*). In agricultural areas, however, aeolian processes are often erosive, as fertile top soil is highly susceptible to wind erosion (*Nanney, Fryrear & Zobeck, 1993*). Therefore, an in-depth understanding of the physical processes of wind-driven sediment transport is critically important.

It is widely recognized that aeolian sediment transport is highly variable in space and time (*Baas & Sherman, 2005*; *Ellis et al., 2012*). Despite our detailed understanding of the physics of wind blown sand (e.g., *Bagnold, 1941*; *Kok et al., 2012*; *Pähtz et al., 2013*), accurately quantifying aeolian sediment patterns remains a challenge. Measurements of aeolian sediment budgets might improve our understanding, but often have limited spatial and temporal resolution. Approaches used to measure aeolian sediment transport include passive sediment traps (*Rasmussen & Mikkelsen, 1998*; *Dong, Sun & Zhao, 2004*; *Sterk & Raats, 1996*; *Basaran et al., 2011*; *Mendez, Funk & Buschiazzo, 2011*), active samplers such as acoustic samplers (*Spaan & van den Abeele, 1991*; *Yurk, Hansen & Hazle, 2013*; *Ellis, Morrison & Priest, 2009*; *Schönfeldt, 2012*), laser particle counters (*Hugenholtz & Barchyn, 2011b*; *Sherman et al., 2011*; *Hugenholtz & Barchyn, 2011a*; *Li, Sherman & Ellis, 2011*), piezoelectric samplers (*Baas, 2004*; *Stout, 1998*), pressure sensitive samplers (*Ridge et al., 2011*) and terrestrial laser scanners (*Nield & Wiggs, 2011*). The physics of wind blown sand are often studied in the controlled environment of a wind tunnel (e.g., *Youssef et al., 2008*; *Van Pelt, Peters & Visser, 2009*; *Goossens, Offer & London, 2000*; *Butterfield, 1999*) but also directly in the field (e.g., *Namikas, 2003*; *Ellis et al., 2012*). However, results from wind tunnel studies cannot be directly translated into field situations, due to differences in turbulence spectrum, wind profile above the bed and variability in environmental factors such as surface moisture, wind direction and velocity, bed elevation, vegetation, sediment composition, lag deposits, surface crusts and fetch. Despite recent progress in rapid data acquisition, where aeolian sediment flux data are collected at high temporal resolution, passive sediment catchers are still frequently used to study aeolian sediment flux.

Passive sediment traps consist of various compartments located at different elevations. Sediment captured within these compartments, provides valuable information about the vertical sediment flux distribution (*Ni, Li & Mendoza, 2003*; *Dong et al., 2003*; *Butterfield, 1999*), which is frequently used to estimate total sediment transport (*Sterk, Herrmann & Bationo, 1996*; *Sterk et al., 2012*; *Sterk & Spaan, 1997*; *Visser, Sterk & Ribolzi, 2004*). Aeolian mass fluxes are quantified by applying non-linear curve fitting through the sediment measurements within the different compartments. However, passive sediment traps have some inherent uncertainties (average of 10%) depending on the specific distribution of sediment within the compartments and their elevation above the surface; whereas sediment mass, inlet diameter, vertical position of the catchers, trapping efficiency, horizontal spacing between catcher arrays and wind direction were also identified as potential sources of error (*Tidjani et al., 2011*). Moreover, variations in elevation from the lowest compartment to the ground (referred to as base elevation hereafter) may also change during the

**Table 1 Type, number and temporal resolution of instruments used during the field experiment on Ameland.** The spatial distribution is shown in Fig. 1B.

| Instrument | Number | Use | Temporal resolution |
|---|---|---|---|
| Anemometers | 3 | Wind velocity profile | 1 min |
| Windvane | 1 | Wind direction | 1 min |
| Tipping bucket | 1 | Amount of rainfall | 1 min |
| Saltiphones | 2 | Transport intensity | 1 min |
| MWAC's | 18 | Sediment flux | Event |

experimental measurement. The vertical distribution of the aeolian mass flux is also important here. When the largest fraction of sediment is transported close to the surface, uncertainties related to the lowest compartment become more important for estimation of total flux; even though vertical flux distribution might also vary through time.

The aim of this study was to characterize aeolian mass flux from wind tunnel and field data by comparing passive trap and high-frequency saltiphone data. Uncertainties caused by the distribution of the different trapping compartments and the influence of the base elevation were analysed for both wind tunnel and field situations. This paper will firstly examine uncertainties resulting from the distribution of different sediment trapping compartments and the influence of base elevation on fast-temporal data acquisition from saltiphones applied in a wind tunnel study. Whereas passive sediment traps collect sediment transport data for each experiment, fast-temporal data is provided continuously throughout the experiment. The latter providing a more detailed analysis of sediment flux. Secondly, the paper will explore the implications of our findings to field studies by testing data gathered from two published studies and a newly acquired dataset. The newer dataset, which utilized three customized sediment catchers, was also used to investigate the variability in vertical sediment flux and total sediment transport.

## MATERIALS AND METHODS

### Data collection

Specific details about the locations and data collection methods used in the two published field studies can be found in *Farrell et al. (2012)* (Table 1) and *Visser, Sterk & Snepvangers (2004)*. Results and data from the wind tunnel study can be found in *Poortinga et al. (2013a)* and *Poortinga et al. (2013b)*, respectively. The data collection procedure used for the new dataset is presented below, while the data of the present study and *Visser, Sterk & Snepvangers (2004)* can be obtained from http://dx.doi.org/10.1016/j.envsoft.2003.12.010.

### Study area

The research took place from November to December 2010 on a beach at the northwestern end of Ameland, one of the West Frisian barrier islands located in the northern extremity of The Netherlands (Fig. 1A). The site is characterized by strong wind and wave dynamics in constructing bedforms and embryonic dune development. Human influence on this part of the beach is minimal compared to the middle section of the island. The

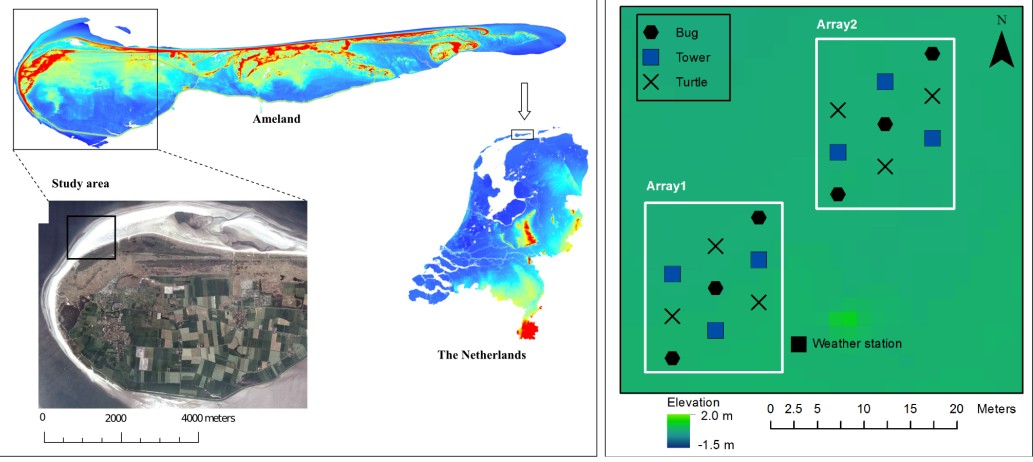

(A) Location of the study area    (B) Field experimental plots (Array 1 and 2)

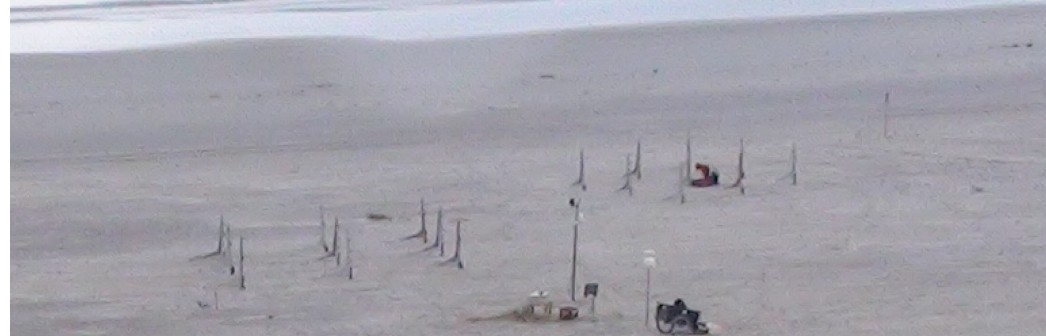

(c) Photograph showing experimental setup

**Figure 1** (A) Location of study area in northern Netherlands, on the island of Ameland and in the aerial photo of the western portion of Ameland. (B) Field experimental setup with specific equipment configuration. (C) Experimental plot at a location along the beach.

study area is located east of a sand bar, which attached to the island in the mid 1980s, causing a progressive, attenuating sand wave to the East (*Cheung, Gerritsen & Cleveringa, 2007*), resulting in relatively wide beaches (>150 m). Figure 1C was taken from the top of the foredune and shows the experimental site at low tide.

## Measurement of sand size

Data was obtained on sediment characteristics, sediment transport and a number of significant meteorological parameters. Surface sands are largely composed of unconsolidated quartz grains with some feldspar and a small fraction of heavy minerals (*Wal, 2000*). To determine sediment size, samples were taken from the beach surface at a number of representative locations across the beach and mixed into one large sample. This sample was dried and sieved in fractions of 50, 100, 250, 500, 1000 and 2000 µm. The median diameter was found to be 180 µm.

### Measurement of sediment flux

Sediment flux was measured using the Modified Wilson and Cook sediment catchers (MWAC). These catchers are designed and used for capturing sediment ranging from dust

to sand. This instrument has been extensively tested in numerous studies (e.g., *Van Pelt, Peters & Visser, 2009*; *Sterk & Raats, 1996*; *Goossens, Offer & London, 2000*; *Poortinga et al., 2013a*; *Youssef et al., 2008*), where efficiencies between 42% and 120% were reported. The original design (*Wilson & Cooke, 1980*) contained six plastic bottles with glass inlets and outlets, placed horizontally at six heights between 0.15 and 1.52 m. These bottles were mounted on a rotating pole with a wind vane. Later studies (e.g., *Sterk & Raats, 1996*) used the same principle, but placed the bottles vertically instead of horizontally (Fig. 2A). Under beach conditions, aeolian sediment transport is governed by saltation, which seldom reaches heights above 15–20 cm. A traditional MWAC sediment catcher would therefore only capture sediment in the lower two or three bottles. This generates significant uncertainty in the analysis, as sediment flux is calculated based solely upon the fitting of an exponential curve through only two or three data points. Therefore, three different designs based upon the traditional MWAC were introduced (Fig. 2), but with all bottles mounted below 25 cm. The first design, nicknamed the "Bug" (Fig. 2B), consists of two stacks of three bottles opposite each other, with their inlets at the same height. The bottles are fixed to a wooden plate with an iron thread to ensure the vertical distance between the two inlets is 5 cm. This design allows for the collection of more measurement points in case of any small horizontal variations in sediment flux, thereby reducing any uncertainty in flux calculations. The second design "Turtle" (Fig. 2C) consists of two bottles on each side, located at various heights. In this design, the bottles are fixed in the original clips, resulting in a larger vertical spacing of 8 cm. In both designs, the horizontal distance between the inlets is 22 cm. The "Tower" design (Fig. 2D) represents the more traditional setup with three or four bottles stacked above each other. The vertical spacing between the bottles is 5 cm, as the bottles are fixed to a wooden plate with iron thread instead of the conventional clips.

To evaluate the differences between the three new designs, they were placed in a $3 \times 3$ grid and separated by a distance of 3 m. After the first event, an additional array of MWACs was installed 8 m from the first array in order to obtain more measurements (Fig. 1B). To ensure careful monitoring of the experiment, this second array was only installed when environmental conditions were favourable. Each array contained three catchers of each type. They were placed in a relatively flat and homogeneous part of the beach to ensure that the measured sediment flux was uniformly distributed. The elevation of the bottles relative to ground level was measured to an accuracy of 1 mm in order to account for changes in base elevation due to ripples; this was done after installation and before removal of the bottles.

### Weather data

A meteorological station with four anemometers was arranged as a vertical array on a tower, which included a wind vane, tipping bucket and two saltiphones (*Spaan & van den Abeele, 1991*), installed on the beach in the middle of the study area (Fig. 1B), recording every minute to a CR10 Campbell datalogger (Table 1) throughout the period of investigation. The on-site meteorological station contained 3 anemometers, measuring wind speed ($ms^{-1}$) at elevations of 0.54, 1.15 and 1.76 m. Pulses from the anemometer

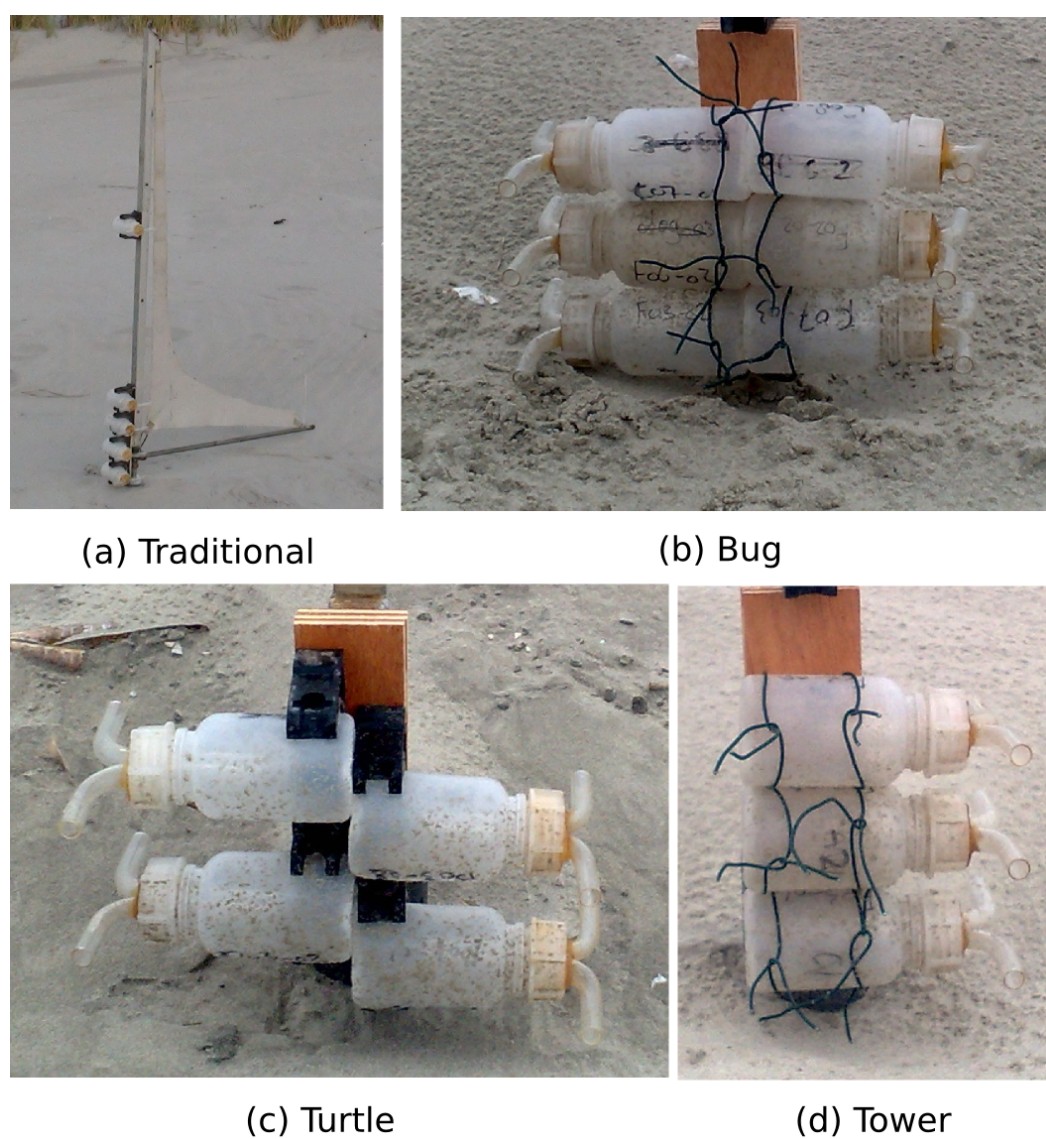

(a) Traditional     (b) Bug

(c) Turtle          (d) Tower

**Figure 2 The traditional MWAC design with the 3 new modified designs.** The traditional design (A) consists of 5 bottles distributed over 1 m. The "Bug" design (B) consists of a total of 6 bottles (3 on each side), the "Turtle" design (C) consists of 4 bottles (2 on each side) in the original clips, while the "Tower" design (D) consists of 3 bottles mounted above each other. Each bottle measures 10 cm high with a diameter of 4.5 cm.

were averaged over the recording period and registered as average wind velocities per minute. Wind direction was measured using the wind vane at a height of 2.5 m, while the tipping bucket recorded rainfall to an accuracy of 0.2 mm.

In order to capture the temporal variability in transport intensity, two saltiphones were placed close to the surface at different locations in the experimental area (Fig. 1B). The saltiphones were also connected to a CR10 datalogger with a digital pulse output signal. For every second, the cumulative number of hits for that second were recorded.

## Data analysis

### *Vertical distribution of aeolian mass flux*

When using passive sediment traps, sediment is trapped in different compartments that are located at different elevations above the surface. Sediment from each compartment was weighed and then plotted against elevation from which a non-linear regression was calculated to estimate total sediment transport. Despite various thoughts on whether to use an exponential, power of five parameter regression curve, the recent literature (*Ellis et al., 2009*) suggests that an exponential decay function (Eq. (1)) is most appropriate to describe aeolian sediment transport.

$$q_z = q_0 e^{-\beta z}. \tag{1}$$

Curve fitting using Eq. (1) enables us to determine the coefficients $q_0$ and $\beta$, also referred to as the portion of creep ($q_0$) and decay ($\beta$), where $z$ (m) represents the elevation and $q_z$ (kg m$^{-2}$) the amount of sediment at elevation $z$. Regression coefficients $q_0$ and $\beta$ can subsequently be used to calculate the total amount of sediment transport $Q$ (kg m$^{-1}$). This is done by the integral of Eq. (1) over the height of the saltation layer (taken as 1 m). Sediment fluxes were expressed in kg m$^{-1}$ rather than kg m$^{-1}$s$^{-1}$, as transport was highly intermittent during some events.

The cumulative transport function (CTF) of an aeolian mass flux ($q_c$) can be described by Eq. (2), using coefficient $\beta$ from Eq. (1). Figure 3 illustrates the relative sediment flux (black line) and the CTF (green line). When studying the characteristics of aeolian sediment flux, the CTF is preferred to the relative sediment flux, as this function is independent of the number of measurement points. Moreover, only coefficient $\beta$ is used in the calculation, and therefore, the shape of the CTF is determined by the specific mass distribution between the different compartments and not by their elevation above the ground.

$$q_c = 1 - e^{-\beta z}. \tag{2}$$

Coefficient $\beta$ (Eq. (1)) can also be used to determine the mean (Eq. (3)), median (Eq. (4)), and lower (Eq. (5)) and upper quartile (Eq. (6)). Figure 3 shows the distribution function as a box-plot (top) and also for the relative sediment flux and CTF (bottom). The difference between the mean (brown line) and median (blue line) is the mean is calculated by the integral and the median by the point where the integral is 0.5. The median splits the CTF into two equal parts, whereas the mean describes the point where the CTF would balance. As the median is less sensitive to outliers compared to the mean, we make use of the median.

$$\bar{q}_z = \frac{1}{\beta} \tag{3}$$

$$q_{z_{50}} = \frac{ln(2)}{\beta} \tag{4}$$

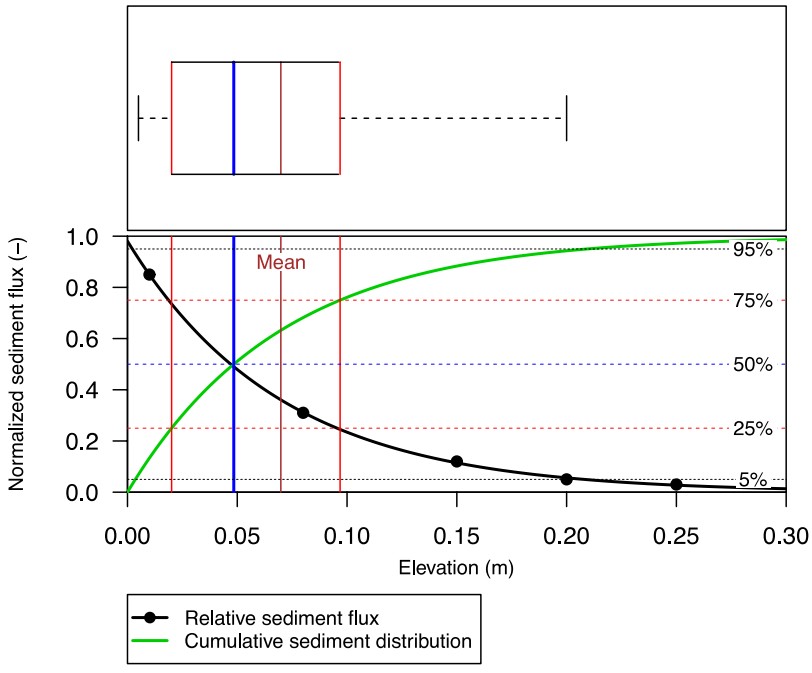

**Figure 3** The vertical distribution of relative aeolian sediment flux (points), the non-linear regression (Eq. (1)) fitted through the data-points and the cumulative sediment distribution calculated from regression coefficient $\beta$. The median (blue, Eq. (4)), mean (brown, Eq. (3)), upper and lower quantile (red, Eqs. (5) and (6)) are also shown as a boxplot.

$$q_{z25} = \frac{ln(\frac{4}{3})}{\beta} \tag{5}$$

$$q_{z75} = \frac{ln(4)}{\beta}. \tag{6}$$

*Dong et al. (2003)* performed a series of wind tunnel experiments to investigate the flux profile of wind-blown sand. They determined the cumulative mass distribution from the measured data. Moreover, they used the equation $q_z = q_0 e^{-b/z}$, where the regression coefficient $\beta$ (here given as $b$) is divided by elevation (Eq. (1)). Regression parameter $\beta$ (as in Eq. (1)) can be calculated by $\beta = 1/b$. The $q_{z50}$ for the different sediment size fractions and wind velocities, using $\beta$, is shown in Fig. 4.

Figure 4 shows the variation in $q_{z50}$ for the various sediment size fractions over a range of wind speeds, especially where coarser sediments are transported at higher elevations, with $q_{z50}$ increasing with wind velocity.

### Uncertainties in estimation of aeolian mass flux

*Ellis et al. (2009)* identified three common methodological inconsistencies and thus sources of uncertainty in measuring aeolian sediment transport using passive traps. These include: (1) inconsistent representation of sediment trap elevations; (2) erroneous or sub-optimal regression analysis; and (3) inadequate or ambiguous bed elevation measurements.

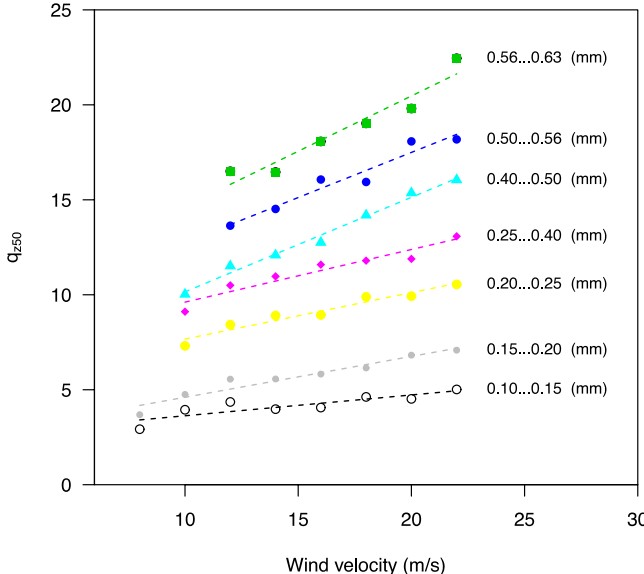

**Figure 4 The $q_{z_{50}}$ for different sediment fractions and wind velocities.** Data were recalculated from *Dong et al. (2003)*.

In addition, the number of trapping compartments and location of the lowest sediment trap are also important considerations. Results from *Dong & Qian (2007)* (Table 1) were used to illustrate how base elevation and number of traps affects sediment flux estimation. They made use of a WITSEG sampler (*Dong, Sun & Zhao, 2004*), which is a vertically integrated wedge-shaped trap with 60 different compartments, where the lowest orifice can be aligned with the surface. The high data density of the WITSEG is advantageous when interested in a detailed description of the vertical mass distribution.

*Dong & Qian (2007)* determined the relative sediment flux (using Eq. (7)), where the relative sediment flux ($qr_z$) at height ($z$) is calculated by dividing the measured sediment flux ($qz$) by the total amount of sediment ($Q$) collected within all compartments. The dimensionless relative height (Zr) was calculated by dividing the actual height ($z$) by the maximum height ($Z$; 0.6 m in their study). After fitting a non-linear regression (Eq. (1)) through the relative sediment flux data, they found a linear correlation between the regression coefficients $q_0$ (portion of creep) and $\beta$ (decay function).

$$qr_z = \frac{qz}{Q}, Zr = \frac{z}{Z}. \tag{7}$$

Figure 5 displays the dimensionless regression coefficients $q_0$ and $\beta$. Using elevation data of the different compartments, we calculated the relative regression coefficient $q_0$ for a sequence of $\beta$'s, while changing the elevation from the base (lines with different colors), but using the same distribution of compartments. Measurements using the WITSEG were taken between 0 and 1 cm, which is in agreement with the experiments. Here, it is important to note the difference in shape between the different base elevation lines. When measurements are taken close to the surface, the correlation between $q_0$ and $\beta$ is almost

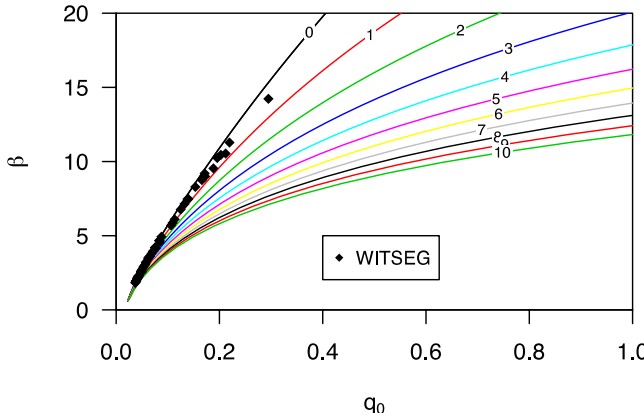

**Figure 5** **The regression coefficients $q_0$ and beta calculated from the relative sediment flux (Eq. (7)) for the WITSEG (data from *Dong & Qian, 2007*; Table 1).** The coloured lines represent the relation between the $q_0$ and beta for different base elevations (shown on the plotted line in cm).

linear, for the domain under consideration. However, when moving further away from the surface, the relationship becomes log-linear (Fig. 5), which has major implications in terms of generating uncertainty in the estimation of $q_0$. Where measurements are taken further away from the surface, a small error in the calculation of $\beta$ has even greater impacts upon estimating $q_0$ compared to measurements taken closer to the surface. An under- or overestimation in the $q_0$ regression parameter can have a significant effect on determining the total mass flux.

The vertical cumulative mass distribution of the aeolian mass flux was investigated for each of the previous studies as well as for the newly collected dataset. The spatial variability for $q_{z50}$ and Q was also investigated for this newly acquired dataset. Due to the relative limited number of datapoints and the desire to maintain the original values in the interpolated maps, we used a simple inverse-distance weighting algorithm, with a minimum of three and a maximum of eight neighbours for spatial interpolation.

## RESULTS AND DISCUSSION

### Wind tunnel data

The wind tunnel study of *Poortinga et al. (2013a)* was used to investigate the CTF of an aeolian saltation cloud (data available in *Poortinga et al., 2013b*). Sediment loss was measured in three different ways using passive sediment catchers, saltiphones and a balance. The normalized sediment flux and CTF, calculated from four saltiphones at various heights above the surface (the highest at 25 cm), are presented in Fig. 6. The BEST, MWAC new and MWAC old represent the three different types of catchers used in the experimental runs. The fit between the non-linear regression line and calculated sediment flux had an average $r^2 = 0.99$, with a minimum $r^2 = 0.96$. The data was divided into high wind velocities (Figs. 6A, 6C, 6E) and low wind velocities (Figs. 6B, 6D, 6F), where the s50, s60 and s80 represent different sediment sizes with a $d50$ of 285, 230 and 170 µm, respectively. At higher wind velocities, more sediment is transported closer to the surface for the s50 and s60 sediment with a $\bar{q}_{z50}$ of 6.71 and 7.94 cm (s50) and 5.17 and 5.61

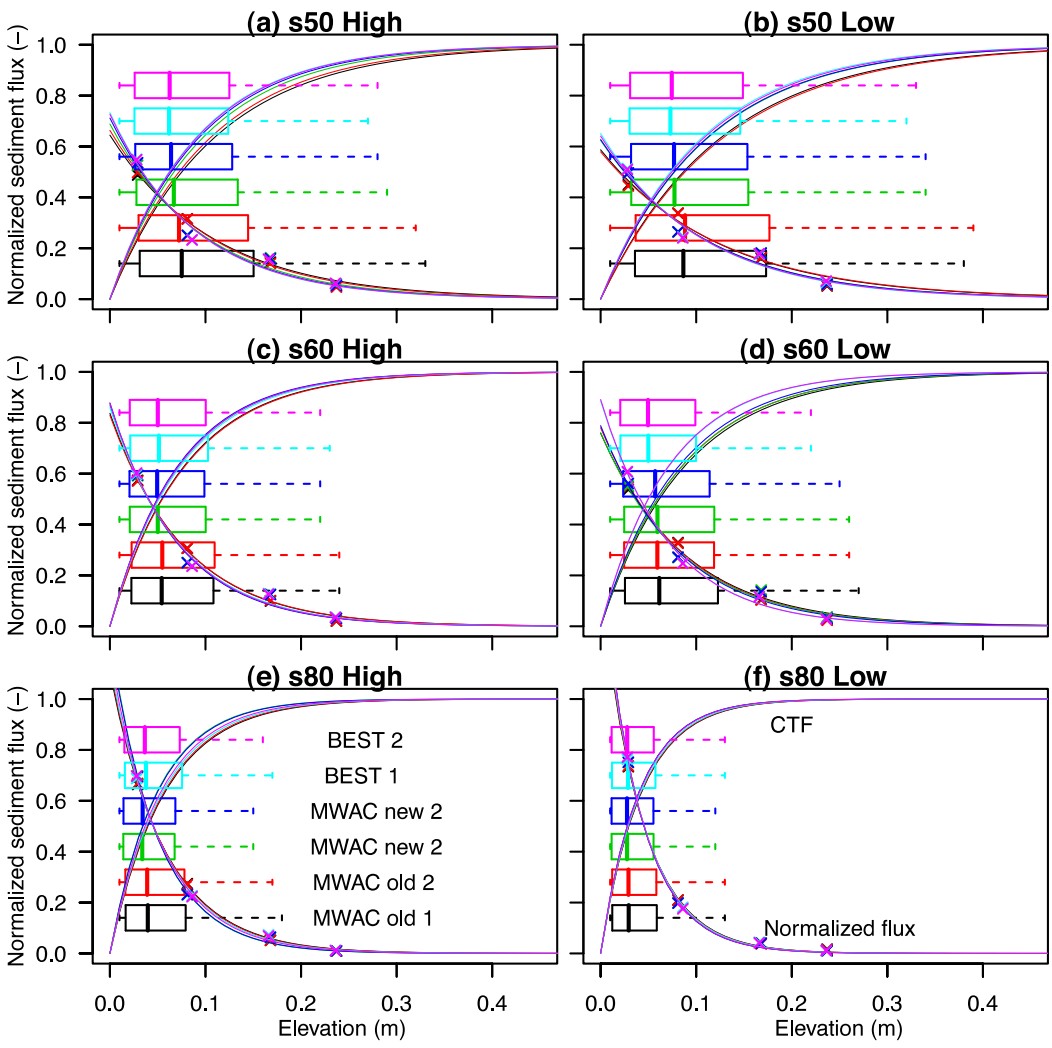

**Figure 6** The relative sediment flux (Eq. (7)) and CTF (Eq. (2)) for three different types of sediment (s50, s60 and s80, with $d_{50}$ 285, 230 and 170 μm, respectively), three different sediment catchers (MWAC old, MWAC new and BEST) and exposed to high (A, C, E) and low (B, D, F) wind velocities. The box plots indicate the median, upper and lower quantile.

(s60) for the high and low wind velocities, respectively. In contrast, the s80 (fine) sediment, $\overline{q_{z50}}$ of 3.68 and 2.84 cm was measured for high and low wind speeds, respectively. The s80 (finer) sediment is transported more readily closer to the surface than coarser sediment fractions (s50 and s60), which is in agreement with the literature (*Farrell et al., 2012*; *Dong et al., 2003*; *Dong & Qian, 2007*).

Saltiphones were also used to rapidly acquire aeolian sediment flux data to enable a detailed investigation of the vertical sediment dynamics. Non-linear regression (Eq. (1)) was applied to the data points, and for all fluxes and data with $R^2 > 0.98$, the $\beta$ was used to calculate $q_{z50}$ (Eq. (4)). In Fig. 7, $q_{z50}$ is plotted against shear velocity for experiments under high (Figs. 7A, 7C, 7E) and low wind velocities (Figs. 7B, 7D, 7F). As shown in Fig. 6, finer sediment has a lower $q_{z50}$ compared to coarser sediment. Despite considerable

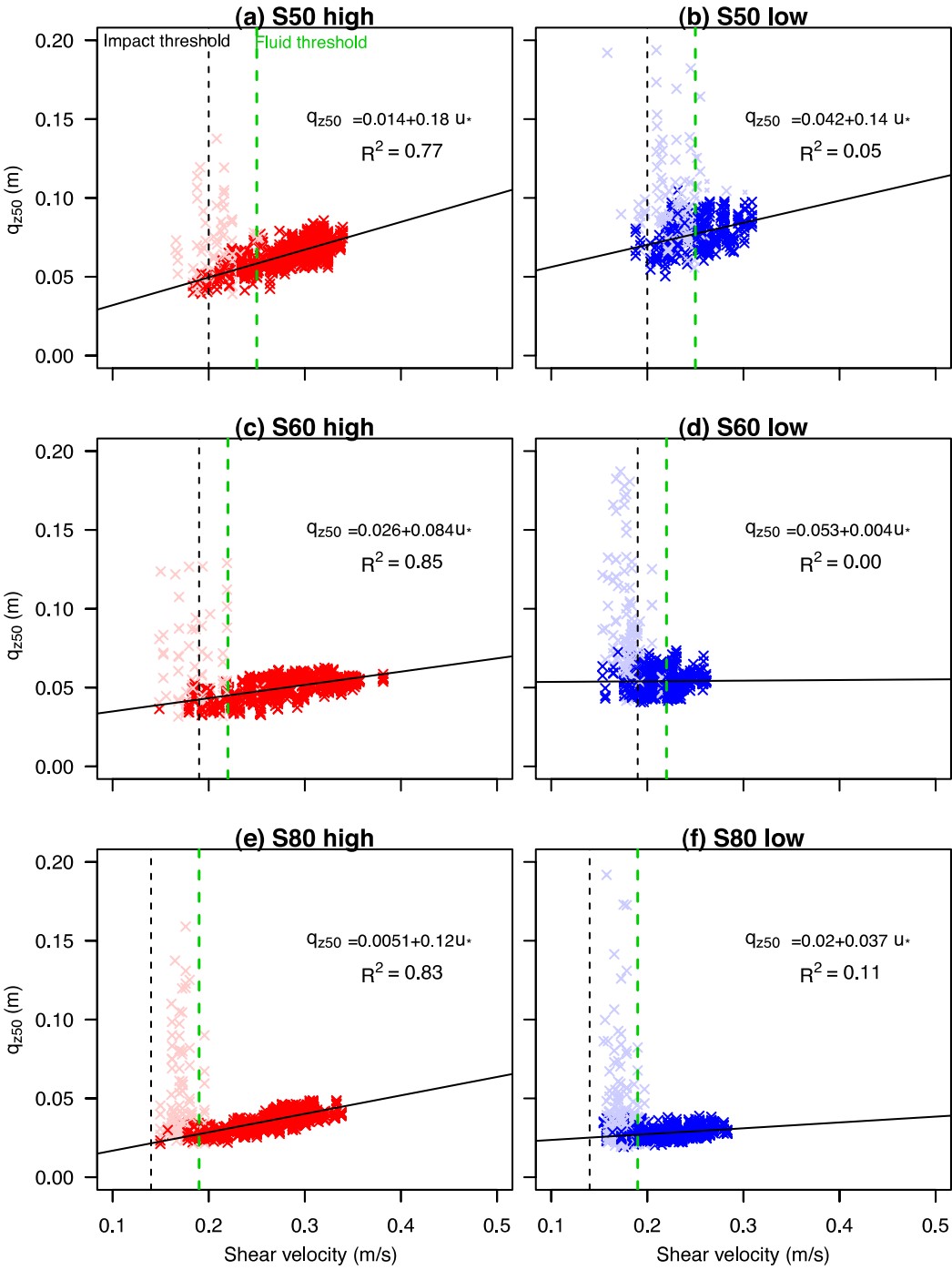

**Figure 7** The shear velocity versus the $q_{z50}$ for three different types of sediment sizes (s50, s60 and s80) under high (A, C, E) and low wind velocities (B, D, F).

scatter, the median $q_{z50}$ values increase with increasing shear velocities. In the region between the impact (vertical black dotted line: Fig. 7) and fluid thresholds (green dotted line), the scatter is considerable. This scatter (indicated with an alpha color), represents measurements with low sediment flux which is more pronounced in low wind velocities.

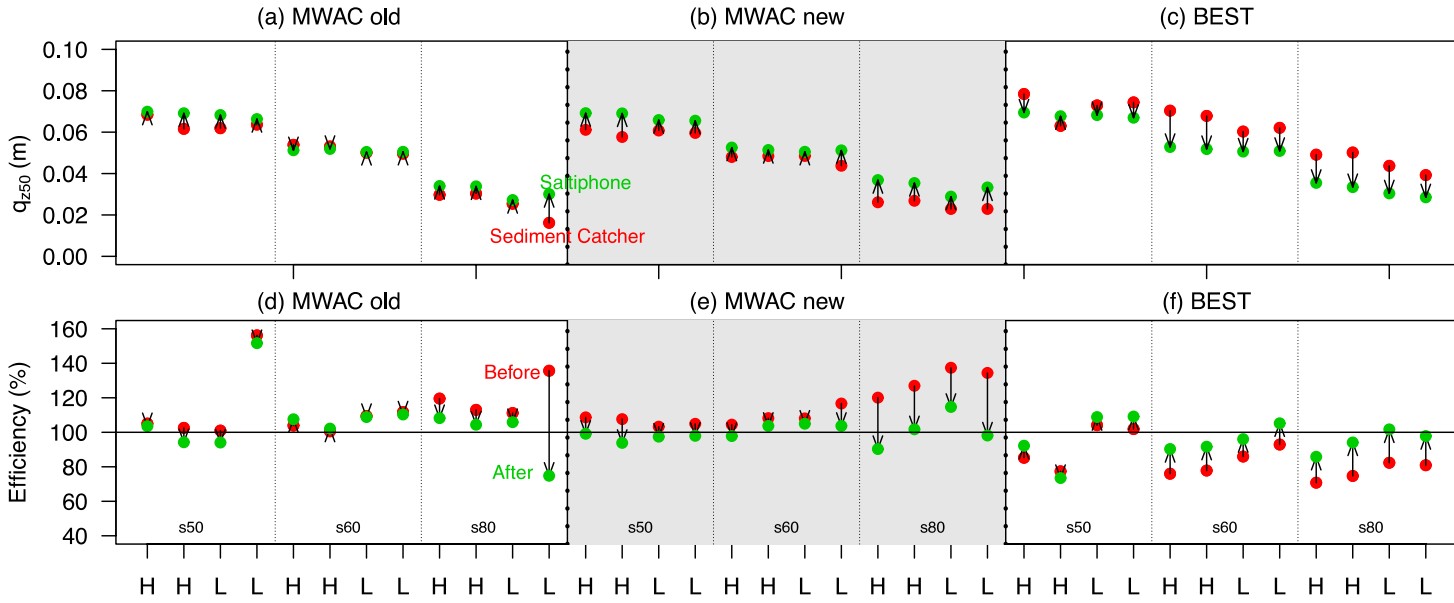

**Figure 8** **The $q_{z50}$ from saltiphones (green) compared with $q_{z50}$ from the sediment catchers (red) (A–C) and the recalculated efficiency (D–F) using the $q_{z50}$ as a reference.** Data is shown for three different sediment catchers (MWAC old, MWAC new and BEST), three different types of sediment (s50, s60 and s80) under high (H) and low (L) wind velocities. During the experiment, the sediment catcher and saltiphones were located next to each other. The arrows (A–C) indicate the shift in $q_{z50}$ used to calculate the new base elevation. The arrows (D–F) indicate the change in efficiency.

A linear regression curve was calculated for the high and low wind shear velocities (straight line). While there is some correlation between shear velocity and $q_{z50}$ under high shear velocities (see $R^2$ of the linear regression in Fig. 7), the $R^2$ under low wind velocities is rather low. All plots show a positive correlation between shear velocity and median $q_{z50}$.

In the wind tunnel experiment, use was made of three different types of sand catchers: the MWAC old, MWAC new and BEST (cf. *Poortinga et al., 2013a*). The findings in Fig. 7 were used to validate the results of these sediment catchers: $q_{z50}$ of the measured sediment flux was calculated for each experiment; and, $q_{z50}$ based on the mean shear velocity during the experiment, was calculated using data from Fig. 7. Figures 8A–8C shows $q_{z50}$ based on the values of the sediment catchers (before) and the values calculated from the saltiphones data (after). Differences between the two values increase from coarse to finer sediment. For measurements using BEST, the differences are generally larger than the other catchers (Fig. 8).

To test whether $q_{z50}$ calculated from the saltiphones (Figs. 8A–8C) provides a better approximation of total sediment flux, $q_{z50}$ was used as a reference point to reposition the base elevation; with the difference between the sediment catcher guiding the repositioning of the traps. The sediment flux was recalculated using this new base elevation. New sediment fluxes were then compared with sediment loss measured by a balance. Figures 8D–8F shows the efficiency of the initial sediment flux estimation (red) and the newly calculated sediment flux (green), where 100% is an exact match with the balance. Some 29 of the 36 measurements were shown to indicate an improvement (Fig. 8). In general, improvements
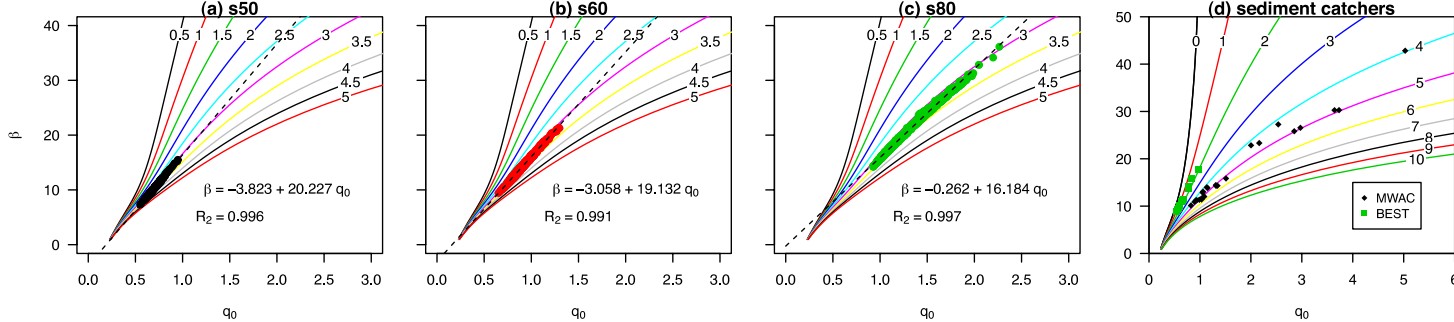

**Figure 9** The $q_0$ and $\beta$, calculated from the relative sediment flux (Eq. (7)) for different sediment sizes (s50, s60, s80) using saltiphones data (graph A–C), and for all measurements using passive sediment traps (graph D). The lines represent different base elevations.

are considerable, and a decrease in efficiency is minimal. For finer sediment, improvements were even higher when compared to coarser sediment.

The relative sediment flux of the saltiphones and sediment catchers were used to determine $q_0$ and $\beta$ (Fig. 9). When the lowest saltiphones were located at 3 cm, we found a strong linear relationship between $q_0$ and $\beta$. Finer sediment had a larger range of regression coefficients, with higher values for $\beta$ given that a higher proportion of sediment is transported closer to the surface. The intercept of the linear regression increases with coarser sediment, whereas the slope of the regression decreases. The difference between the intercept and slope of the s50 and s60 sediment is small. For passive sediment catchers, there is good agreement between the calculated base elevation and the experimental results (Fig. 9). The BEST catcher was located 1.5 cm from the surface, whereas the lowest trap of the MWAC catchers was located between 4 and 5 cm. Mean measurement error was 1.3 mm with a maximum of 2.4 mm.

The disagreement in vertical flux distribution between the saltiphones and sediment catchers, and also between sediment loss measured by the balance and the calculated flux from the sediment catchers, is mainly caused by the specific configuration of the sediment catcher. For instance, when applying an exponential regression function, the elevation, orientation and measurement accuracy of the lowest bottle largely determines the result as finer sediment is more susceptible to errors compared to coarser sediment. Figure 10 presents the experimental outputs when using the MWAC and BEST catchers with s80 (fine) sediment, highlighting the measured relative sediment fluxes (black dots), including the exponential regression (Eq. (1)) and a linear regression. The BEST catcher contains one data-point below the $q_{z50}$ while the MWAC has none. The influence of the lowest data-point is significant, as it determines the intersection with the $y$-axis and thus the total sediment flux. As for fine sediment, errors will be more pronounced as a larger portion of the mass is transported close to the surface, there are small inconsistencies in the orientation of the catcher, and thus measurement issues occur with the elevation or difference in efficiency under different mass flux density. Applying a linear function to the points close to the surface, and a power function for the higher located points (cf. *Poortinga et al., 2013a*), will therefore give more coherent results, as the effect of the lowest point on the total mass flux is reduced. Moreover, *Ni, Li & Mendoza (2003)* showed that saltating

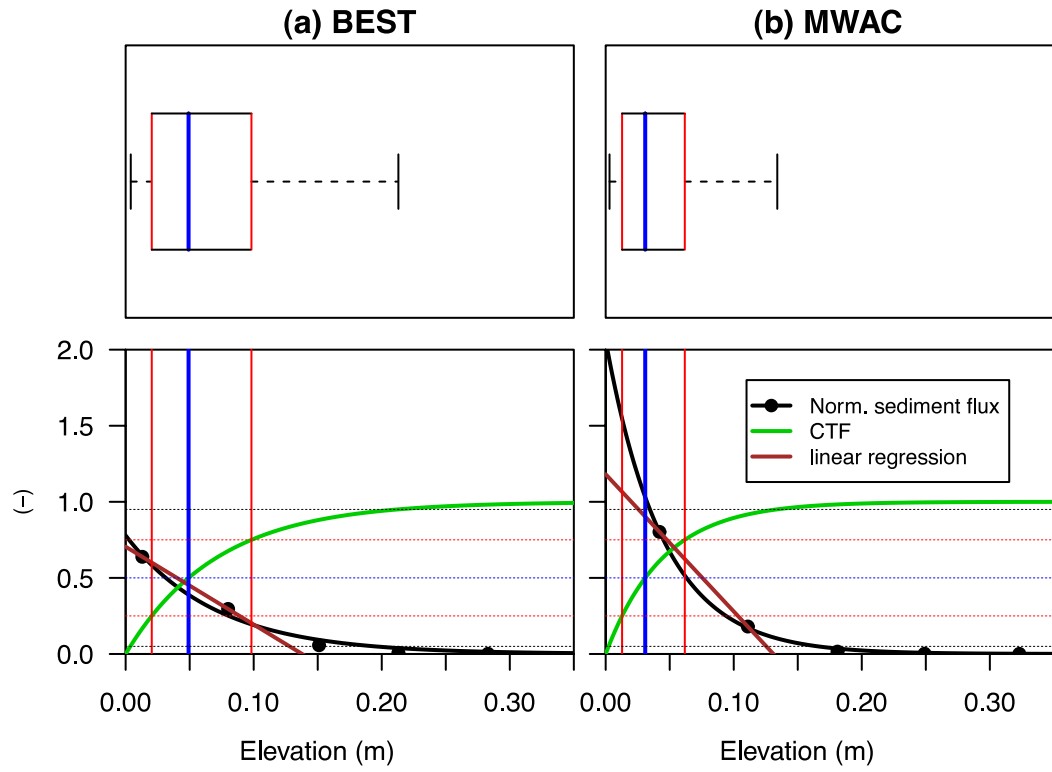

**Figure 10 The vertical distribution of the relative aeolian sediment flux for the BEST and MWAC sediment catchers.** The dots identify individual measurements for the s80 sediment size, while the non-linear regression curve is shown in black and the CTF in green. Furthermore, a linear function was plotted through the two points located closest to the surface (brown). The $q_{z50}$ (blue: Eq. (4)), $\bar{q}$ (brown: Eq. (3)), upper and lower quantile (red: Eqs. (5) and (6)) are also shown as a boxplot.

grains follow an exponential decay function, whereas creeping and reptating grains deviate from it. The mathematical description might therefore also be a source of uncertainty.

### Field data

Where wind tunnel studies are limited in the replication of complex turbulent wind structures as seen in the field, field studies do not have the advantage of a controlled environment where specific parameters can be fixed. Surface moisture and bedform development, for instance, are known as important limiting factors in sediment transport, and can negatively affect measurements. Data from *Farrell et al. (2012)* were used in a re-analysis because their short-lived experiments contained several data points close to the surface. For the sub-environment Cow Splat Flat Fine (CSFF), $q_{z50}$ were arranged according to date (Fig. 11A) and $q_0$ and $\beta$ were calculated for the relative flux (Fig. 11B). For this sub-environment, the calculated elevation from the surface strongly agreed with the measured values. The variation in $q_{z50}$ was best explained when arranging them according to measurement date; where no relation was found with shear velocity (ranging from 0.45 to 0.54 ms$^{-1}$) or grain size. A logical explanation would be the effects of surface characteristics such as surface moisture and incipient bedform development. However, this is far from conclusive, as it was also found that $q_{z50}$ increased with decreasing

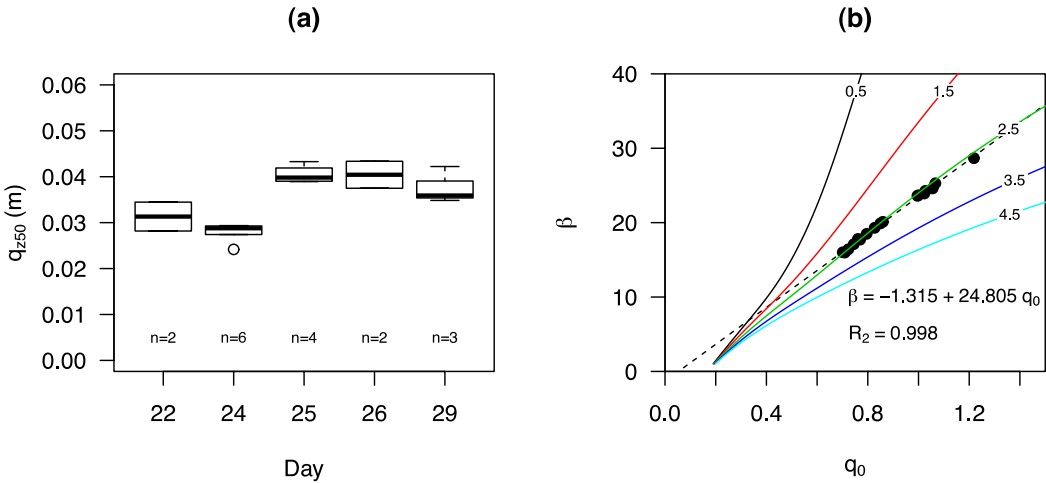

**Figure 11 The $q_{z50}$ for experiments performed (at the sub-environment Cow Splat Flat Fine (CSFF)) on different days (A) and the $q_0$ and $\beta$ for all events combined (B).** Coloured lines represent different base elevation. Data was obtained from *Farrell et al. (2012)*.

$R^2$ (ranging from 0.968 to 0.999). The same study also took three measurements at the beach sub-environment over two consecutive days. These measurements received specific attention, as they were taken at a wet and immobile foreshore without visible bedform deformation. We found $q_{z50}$ values of 3.7, 4.5 and 3.1 cm with $R^2$ of 0.966, 0.890 and 0.997, respectively. As this dataset only contains three data-points with varying $R^2$, it is difficult to draw conclusions based on $q_{z50}$ or measured base elevation.

*Visser, Sterk & Snepvangers (2004)* conducted experiments on three different geomorphic units: degraded, valley and dune. Besides sand, the soils in this area also contained considerable quantities of silt: 19.4, 15.9 and 13.0% and clay 21.6, 5.1 and 3% for the degraded, valley and dune site, respectively. The study obtained results for 11 different events in the year 2001, with 17 MWAC catchers installed at each site. In order to remove uncertainty from the data while maintaining an acceptable number of data points, measurements with $R^2 < 0.95$ were removed from the dataset. This differs from previous studies, where an $R^2 < 0.98$ was used as measurements were taken over longer periods. Furthermore, when comparing the different units (Figs. 12A, 12C, 12E), it was found that $q_{z50}$ is highest for the degraded site, followed by the valley and dune site. The degraded and valley site have higher fractions of silt and clay, which are transported over higher elevations. However, surface crusts might also cause saltating particles to reach higher elevations. The variation in $q_{z50}$ within an event is generally low for the degraded and valley site, but slightly higher for the degraded site. The variation between events is also small, except for the events on 10 and 13 July (Fig. 12). Here the values for $q_{z50}$ are high and have a large variation. During these events, large amounts of dust were transported through the study area. No clear relation was found between $q_{z50}$ and wind velocity.

Figures 12B, 12D, 12F shows $q_0$ and $\beta$, calculated from the relative sediment flux. As surface elevation varied for the different measurements, the points are plotted on different curves. Due to a lower decay rate (coefficient $\beta$) at the degraded site, points are still closely

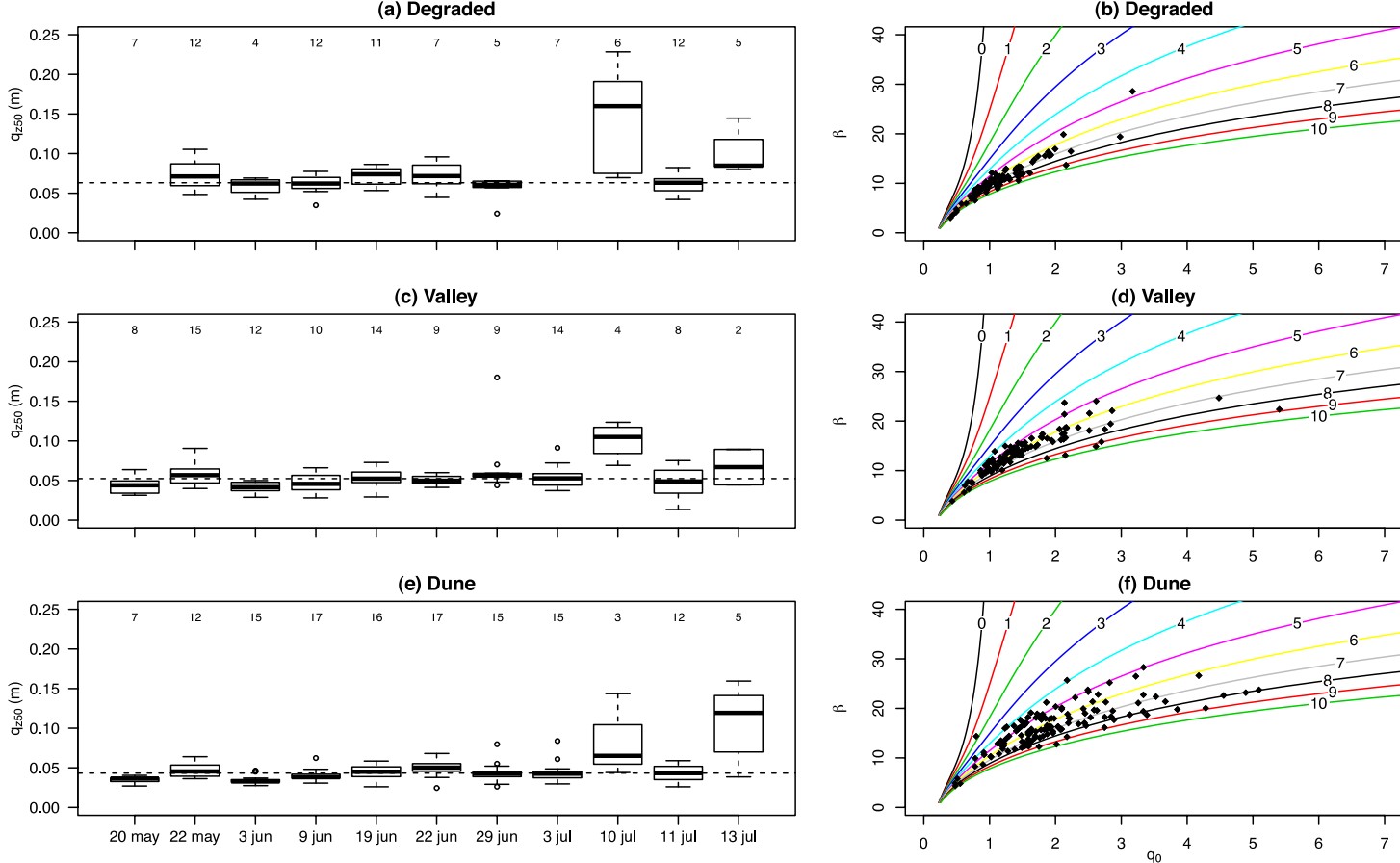

**Figure 12 The $q_{z50}$ for 12 events in three different geomorphic units (A, C, E) and the relation between $q_0$ and $\beta$ (B, D, F), where the lines represent different base elevations.** Data was obtained from *Visser, Sterk & Snepvangers (2004)*. Numbers indicate the number of measurements included.

related. However, at the dune site, sediment is transported closer to the surface, resulting in higher decay rates. As the lines spread with higher decay rates, there is higher spread in points. Compared to the degraded site, there is larger uncertainty in $q_0$ for the dune site, as small errors in $\beta$ will result in larger errors in $q_0$ (Fig. 12).

Figure 13 displays the uncertainty in $q_0$ for one event (May 22) at the dune site. Here, the measured elevation is shown in red; where elevation is based on the relative $q_o$ and $\beta$, shown in green. For this event, all calculated elevations are lower compared to the measured. This indicates that there is a likely error in the measured base elevation, leading to an overestimation of $q_0$. As expected, errors in $q_0$ are largest for the dune site, followed by the valley and degraded site. However, the larger error in $q_0$ does not directly correspond to a larger error in base elevation. Elevation here was estimated using $z = ln(q_z/q_0)/\beta$ (Eq. (1)), with the higher decay rates at the dune site having a more pronounced effect on Eq. (1) than the larger range of $q_0$. For flux estimation, on the other hand, small changes in elevation have a much larger impact, as a greater portion of transport takes place close to the surface.

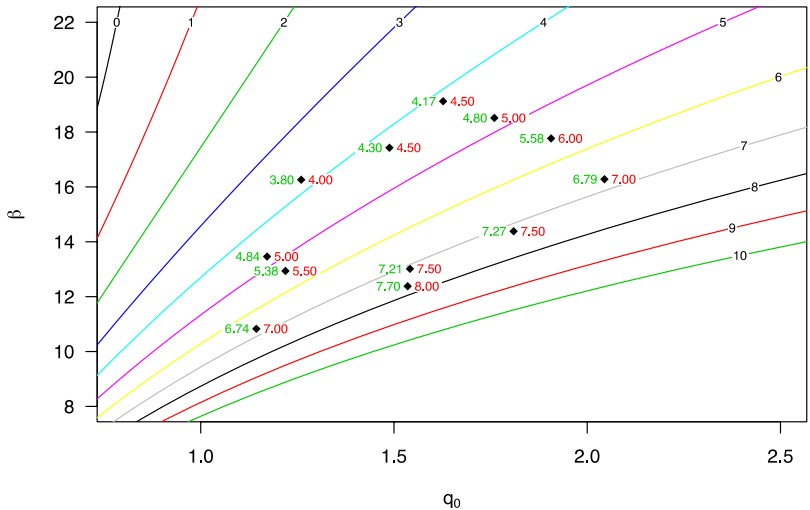

**Figure 13 The $q_0$ and $\beta$ calculated from the relative sediment flux for the Dune site (May 22).** The lines represent different base elevations. The red numbers are the measured elevations whereas green values are the calculated elevations based on relative $q_0$ and $\beta$.

### The field experiment

New data were collected for six different events (Fig. 14). The duration of the experiments varied from a few hours to two days, whereas saltation was measured for several hours. The averaged shear velocity during these saltation periods varied between 0.30 ms$^{-1}$ for event 1 and 0.41 ms$^{-1}$ for event 3. Wind directions predominantly came from the E-NE while only event 3 had variable wind conditions (Fig. 14). During events 2, 3 and 4, rainfall was recorded. Saltiphone data were also included as an indication of the degree of saltation activity.

Of the three catchers (Turtle, Bug and Tower) used in the experiments, two of them contained compartments on both sides of the catchers. The impact of this horizontal variation on sediment flux was investigated by evaluating $R^2$ as a non-linear regression (Eq. (1)) was applied to all measurements with at least four data-points. For the bug sampler, a non-linear regression was applied to both sides of the catcher, where the middle bottle of the opposite side was included. The results (Fig. 15A) show that the Tower has the best correlation, followed by the Turtle. The Bug has the poorest performance but contains more measurements. Disturbance of the airflow might have caused the decrease in performance. In general, most measurements have a very high correlation, indicating only minor impacts of horizontal variability. However, to exclude the effect of horizontal variability and other sources of uncertainty, only measurements with an $R^2 > 0.98$ were included for further analysis.

The $q_{z_{50}}$ values are shown in Fig. 15B. Values for event 6 are higher compared to other events, which is most likely caused by the frozen surface. During events experiencing rainfall, sediment was generally transported over higher elevations. However, the configuration of the traps on the catcher were also found to have an impact. Figure 15C shows that the the Turtle design gives generally higher values for $q_{z_{50}}$ compared to the

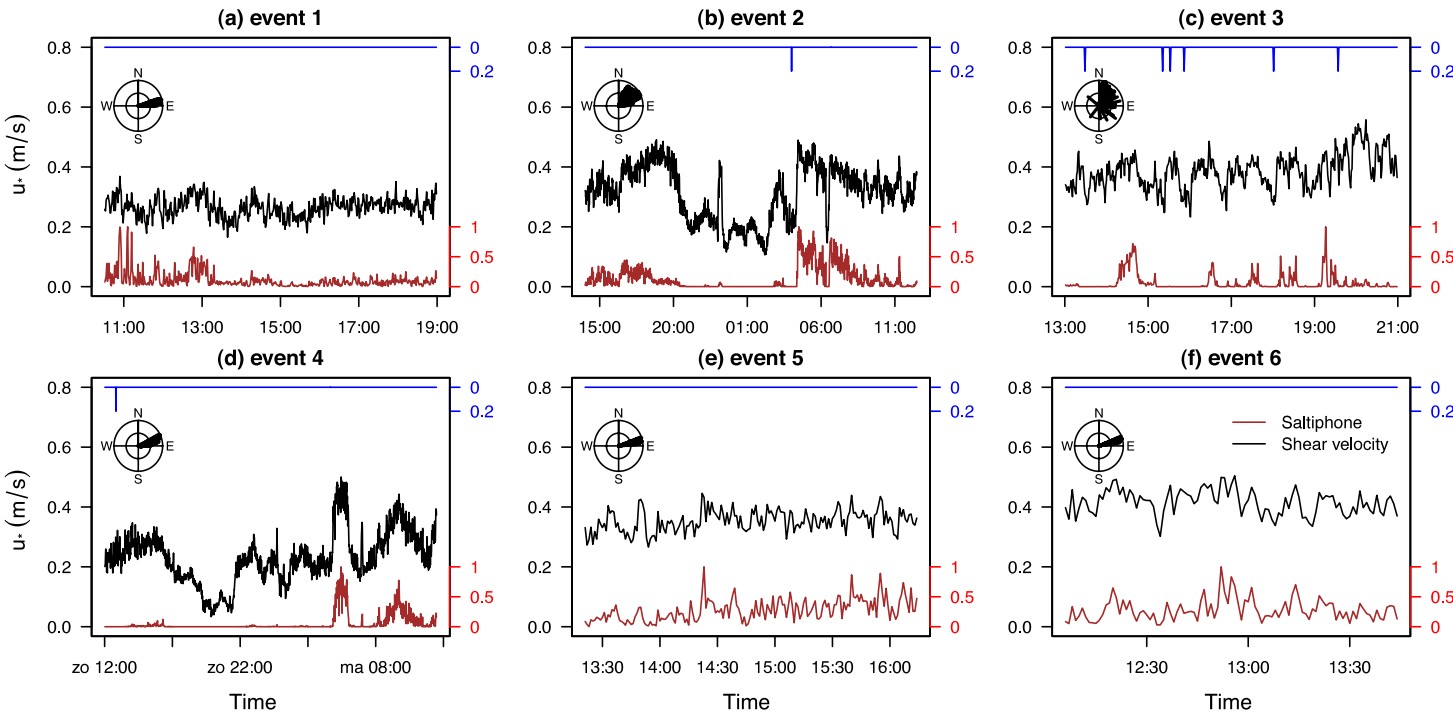

**Figure 14 Shear velocity, wind direction, rainfall and normalized saltation activity for six events.** The normalized saltation activity was determined using the total count of the saltiphones, divided by the maximum count during the event.

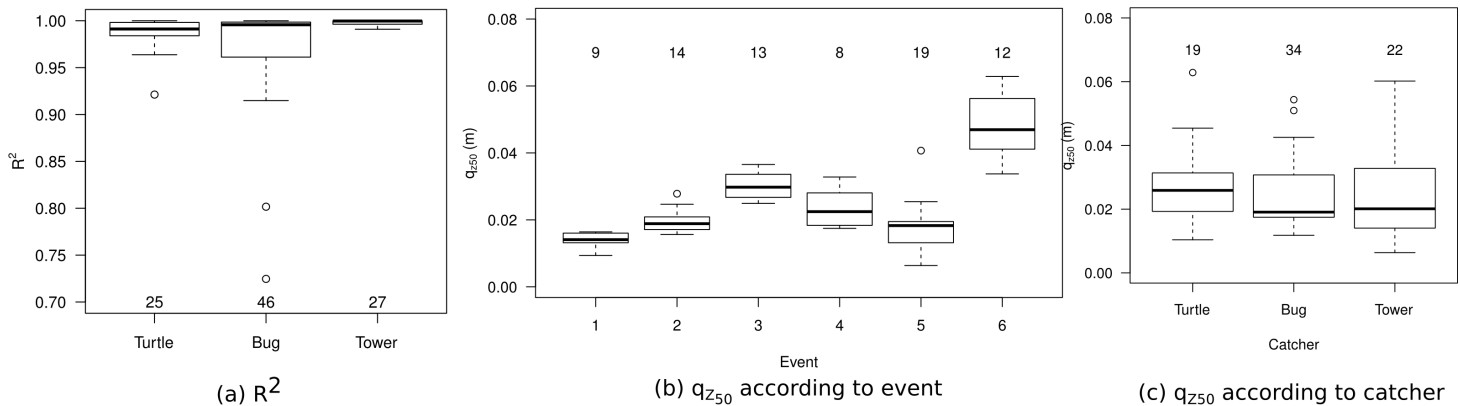

**Figure 15  (A) The $R^2$ for the three sediment catchers, (B) the $q_{z50}$ for every event and (C) the $q_{z50}$ according to catcher.** Numbers indicate the number of measurements included.

other two designs. This is caused by point density being close to the surface. For the Bug and Tower designs, regression coefficient $\beta$ is based on one point close to the surface, whereas the Turtle has two data points. The range in $q_{z50}$ is larger for the Tower compared to the Bug, as the Bug has two data points at approximately the same elevation, with the measurement being refuted when these points do not match. Measured base elevation was in good agreement with the calculated base elevation, with an average difference of 0.7 mm

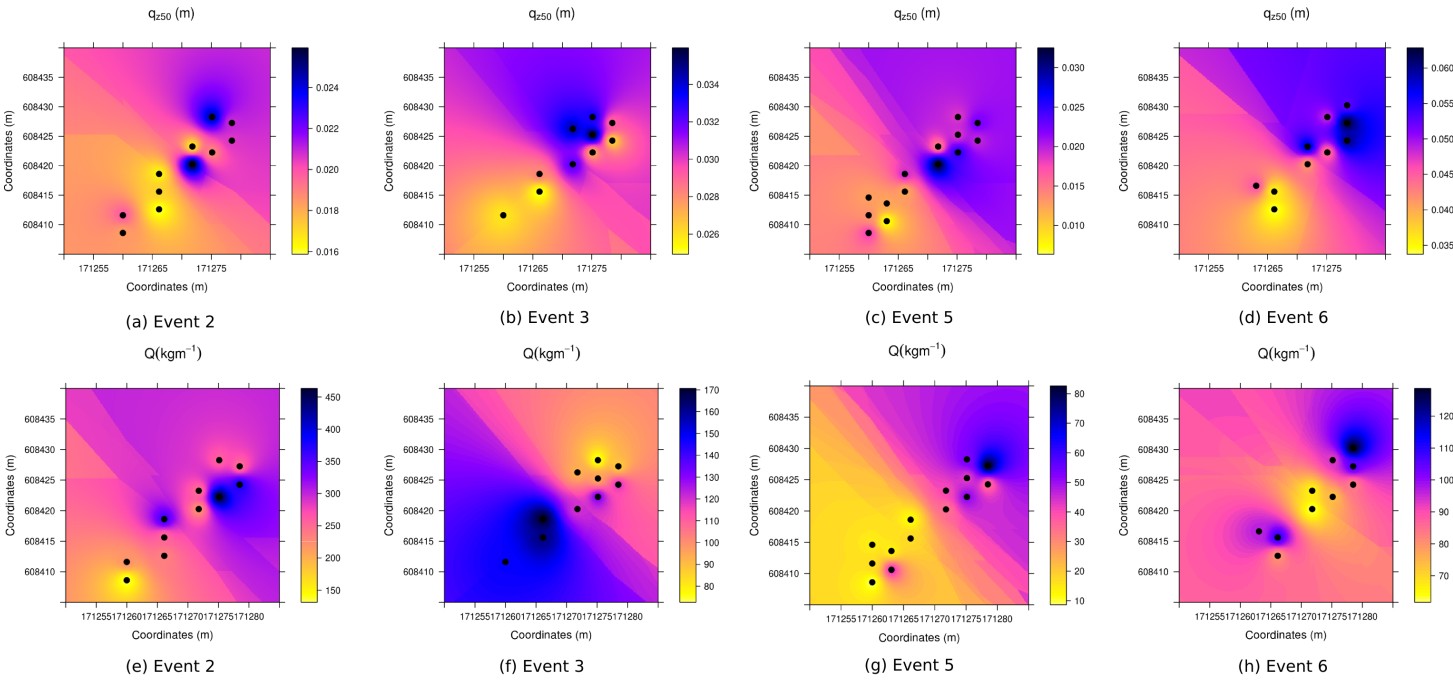

**Figure 16** The spatial distribution of $q_{z50}$ (A–D) and $Q$ (E–H).

and a maximum of 5 mm; with the Turtle displaying the largest variation, followed by the Bug and the Tower.

An Inverse Distance Weighting (IDW) algorithm was used to investigate the spatial variability of $q_{z50}$ and $Q$ (Fig. 16). We selected events 2, 3, 5 and 6, as during these experiments, two arrays of catchers were used. For all events, the lower-located array has lower values for $q_{z50}$ compared to the higher array. Based on our observations, we can confirm that the surface of the upper array was generally wetter than the lower array. This is in line with other findings by *Nield & Wiggs (2011)*. *Farrell et al. (2012)* also found that sediment is transported over higher elevations on wet surfaces. The spatial variability in saltation height (and thus surface characteristics) shows no alignment with the total transported sediment. Furthermore, the large variability in sediment flux between the different events, suggests there is also large variability in total sediment transport within individual events. Peak values are eight times higher than the lowest values within measurement plots. In general, there is good agreement in measured sediment flux between points located close to each other. However, within meters of these measurements we can see major differences in total sediment flux. Besides the limiting effect of surface moisture on aeolian sediment transport (*Namikas & Sherman, 1995*; *Cornelis & Gabriels, 2003*; *Neuman, 2003*), the variability in sediment flux can be attributed to the presence of aeolian streamers (*Baas & Sherman, 2005*; *Baas, 2008*) and/or fetch length (*Davidson-Arnott, MacQuarrie & Aagaard, 2005*; *Bauer et al., 2009*; *Delgado-Fernandez, 2010*).

Differences in vertical sediment flux as found in the wind tunnel studies have limited validity for field studies, as surface conditions were found to have an important impact on saltation. Wet, frozen or crusted surfaces increase saltation height, as particles retain a higher proportion of their impact energy (*Farrell et al., 2012*). This effect was regarded as localized due to the spatial variability of the surface. Moreover, saltation trajectories have a scattered pattern between impact and fluid threshold. This may impact results from the field, as during some events, transport was highly intermittent due to fluctuations in wind speed (*Davidson-Arnott & Bauer, 2009*; *Stout & Zobeck, 1997*). However, additional rapidly-acquired field data are necessary to study this phenomena in more detail.

## CONCLUSION AND RECOMMENDATIONS

Using fast-temporal data on aeolian sediment transport in a wind tunnel, we found that $q_{z_{50}}$ displays a scattered pattern between the impact and fluid threshold, but shows a linear increase with shear velocities above the fluid threshold. Furthermore, it was shown that errors that originate from the distribution of compartments and the location of the lowest sediment trap can be identified using relative sediment flux. In field situations, shear velocity was not found to be the most important controlling factor in vertical sediment flux characterization. Instead, surface moisture was an important control, although particle characteristics of the source area should also be considered. Errors have a more pronounced effect on sediment flux estimation for fine compared to coarse sediment, as fine sediment fractions have a larger portion transported closer to the surface. In order to reduce uncertainty, it is recommended to locate multiple traps closer to the surface.

## ACKNOWLEDGEMENTS

We would like to acknowledge the following people who have made the completion of this manuscript possible. Pierre Jongerius and Corjan Nolet, thank you for all your hard work in assisting us with the field data collection. Three anonymous reviewers are also thanked for their invaluable comments.

### Funding

The study was paid by the Soil Physics and Land Management Group. The funders had no role in study design, data collection and analysis, decision to publish, or preparation of the manuscript.

### Grant Disclosures

The following grant information was disclosed by the authors:
Soil Physics and Land Management Group.

### Competing Interests

The authors declare there are no competing interests.

## Author Contributions

- Ate Poortinga conceived and designed the experiments, performed the experiments, analyzed the data, contributed reagents/materials/analysis tools, wrote the paper, prepared figures and/or tables.
- Joep G.S. Keijsers and Saskia M. Visser conceived and designed the experiments, performed the experiments, analyzed the data, contributed reagents/materials/analysis tools, wrote the paper, reviewed drafts of the paper.
- Jerry Maroulis analyzed the data, contributed reagents/materials/analysis tools, wrote the paper, reviewed drafts of the paper.

## Data Deposition

The following information was supplied regarding the deposition of related data:

3tu datacenter (http://datacentrum.3tu.nl/en/home/): DOI 10.4121/uuid:e1c16aac-02e2-4ec9-b2aa-b171cb034293.

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
