# Peer review of "Measurement uncertainties in quantifying aeolian mass flux: evidence from wind tunnel and field site data"

_PeerJ, doi:10.7717/peerj.454_

## Round 0.1 · original submission · Major Revisions

All three reviewers find merit in the manuscript, but highlight major revisions. Please respond positively to the critical appraisals and the suggestions of reviewer one for increasing the utility of the manuscript for the modelling community.

Reviewer 1 ·

Basic reporting

The article is an interesting approach to examine saltation layer dynamics.
However the paper lacks attention to detail in formatting and content throughout. These issues take from the strength of the research. These have been highlighted in the .pdf attached. I would highly recommend that the authors proof read their papers before submitting them to any journal.
The authors examined the height of the saltation layer (using the median height as the surrogate. Ultimately, it is not clear what new information is provided or the usefulness of some of their approaches (e.g. CDF). The background section does not address many important themes in aeolian literature e.g. findings from previous studies using passive traps on controls of decay? The authors can get to the results much quicker. It is known that traps closest to the ground are most important as the vertical distribution decays rapidly away from the bed. Many wind tunnels studies have found this and also targeted the impact of wind velocity and sediment size of decay rates. Field studies are ambiguous - and, as the authors recognize, many of the problems stem from lacks of control of the surface and sediment. This in essence is the argument of the authors. The strength of their paper is their own field data and this should be the focus.

Experimental design

Please see comments in attached .pdf.
The article is not up to standard for this scientific journal. It requires major revisions and much more attention to detail. The links with the literature and previous experiments are weak and the paper fails to provide the necessary literature review.
The figures are not explained or useful in many cases (Fig 15) or illegible in others (Fig 1).

Validity of the findings

The conclusions are not new. The authors use existing data for the most part but do contribute with their own field based instrumented study. If the paper provides their own data in a table format that other researchers can use then it would add value to the paper and worthy to be published.

Annotated reviews are not available for download in order to protect the identity of reviewers who chose to remain anonymous.

Reviewer 2 ·

Basic reporting

No major conflications to the journal standard

Experimental design

1. I personally doubt the efficiency of the MWAC to capture the sand, especially in the field for hours. Have the authors tested it in the wind tunnel for similar duration? This may partially explain the disagreement in vertical flux distribution between the saltiphones and sediment catchers. Also I am not quite sure if the sensitivities between saltiphone sensors are similar. Previously we have known this issue in the old saltiphone model, but I do not quite sure if the new model have overcome this problem. Even if the sensitivities are the same, the saltiphones closer to the bed may be saturated with more intense saltation, and also you may have more counts in the upper saltiphones because stronger sand momentum. Anyway, it is very dangerous to use particle counters to derive the vertical flux profile.

2. I am not quite sure how did you define the base elevation. According to Ellis, (2009), the base elevation should be the elevation of zero, but from the discussions of fig 9, it seems the authors believes it is the elevation of the bottom trap. If it is the latter, it should be clearly defined.

3. Why choose IDW, not other interpolation method?

Validity of the findings

1. For the discussion of variability. on fig 15, there should be 18 MWACs , however, I only see 10-14 dots. Why?

Additional comments

I value the effort of the hard work of the field explorations. However, I think there are some major flaws of this paper, please address my comments above. Also I have the following minor comments:
Figure 1, right image and right image should be switched.
Line 205 , should be 170 μm
Line 207-209, does not make sense to me.
Figure 6, what are those boxes in the fig

Reviewer 3 ·

Basic reporting

1) My main criticism is that I think the study could use some focusing (some of the comments below cover this as well). There are many results which I think would be interesting to the aeolian community, but someone reading the present version of the manuscript won’t necessarily catch them. Perhaps explain to the readers explicitly why each set of results matters or could be used to test models, develop better predictions, develop better measurements, etc.

2) I think the applicability of the results for modelers could be improved with some small changes to the presentation. Perhaps plot the different environments by median grainsize and sediment flux to give a general idea of the variability in q~ that could be useful for modelers. While this study focuses quite a bit on q~, what about q75 and q25? This would also be useful for modelers (e.g., references in Kok et al., 2012).

3) The introduction could be improved. Include a paragraph that lists the key questions addressed by the study, then address each question in the subsequent studies. This would help readers grasp the aims of the study and sell the work a bit more.

4) I found the introduction could be more pointed in describing the importance of vertical profiles of flux within the saltation cloud. Is this to better quantify sediment flux (e.g., better measurement protocols)? Or, general information for modelers to test their models with? Or both?

Recommendations: perhaps list what we should be doing in the field to get quality sediment flux measurements. How do we measure the height of the bed? What pattern of MWAC samplers are recommended? etc. Give us some specific direction. I think there is a bad habit of many readers out there of reading the abstract, skimming the figures, and then reading the recommendations. I think making some point form take-home points in the recommendations will allow these readers to get something easily from the paper.

Experimental design

5) The height of the bed is an important problem. From the data presented here, it appears that the profile of sediment flux can be used to back-calculate the location of the traps. Is this a suggestion to do this? If not, how could one determine the elevation of the bed. I think some acknowledgement of the problem of defining a base level is needed – ripples are constantly rolling past these arrays, it really isn’t appropriate to just stick a ruler to the bed immediately below the sensors or traps.

6) The spatial variability section needs significantly more context: photos, measurements, descriptions. What does this place look like? What is the source of variability?

Validity of the findings

The main take-home conclusion is that flux measurement equipment should be placed as low as possible within the saltation cloud. This is supported by the data.

Additional comments

Specific comments:
Abstract: define q with ~ on top. I think it would be better to use z underscore q50 as a symbol (or similar) as q is reminiscent of some sort of sediment flux. What is “relative sediment flux”?

Introduction: I think a paragraph is required further elaborating on the rationale for measuring the vertical profile of sediment flux. While the vertical profile is important, the spatial variability in sediment flux is perhaps a larger issue facing those measuring flux (e.g., Ellis et al., 2011; Baas and Sherman, 2005). How does this study contribute to the extensive literature on vertical profiles of flux?

L31: are these active sensors? I’d consider these sensors passive as they sit there and listen for saltation impacts.

Line 33: perhaps also reference the comment and reply to the Sherman et al., 2011 study. There are errors and important inadequacies in Sherman et al., 2011 that are discussed in more detail in the comment and reply by Barchyn and Hugenholtz.

Lines 37-40: perhaps elaborate here the main differences between wind tunnels and field situations. The big problem lies in the spectrum of turbulence and profile of wind above the bed, beyond environmental variables.

Line 45-46: Detail how flux is estimated commonly, and describe assumptions.

Line 73: spelling: Data were.

Figure 1b: describe shading in the background. Elevation? Provide scale for shading. Change ‘Experiment’ to ‘experiment’.

Figure 1 caption, reverse right and left, or maybe better: just refer to ‘a’ and ‘b’.

Line 86: ‘governed’ might be a better word

Line 108-109: detail uncertainty in elevation from ripples.

Line 117: express velocities in m / s, recorded every minute.

Line 132: Barchyn et al., 2011 didn’t say anything about mass flux distribution above the bed, this was an attempt at improving measurement technology by limiting the calibration work required by the community through standards.

Line 135: usually flux is a rate, e.g., kg /m2 / s (check elsewhere).

Figure 2: zoom into bottles on the far left setup. Put scales on images.

Line 151: change to: ‘less sensitive to outliers’

Figure 3: label y axis

Line 267: perhaps elaborate further on the differences between wind tunnels and field situations. The biggest difference aside from precise environmental control is the temporal spectrum and spatial pattern of turbulence. It is simply impossible to reproduce the size of turbulent structures seen in the field in a wind tunnel. Given that the spectrum of turbulence overlaps with the spectrum of saltation response (Baas, 2006, GRL) the saltation cloud will constantly be lagging behind the wind forcing. Flux responds nonlinearly to shear stress and with impact hysteresis (E.g., fluid vs impact threshold) – thus the spectrum of turbulence will matter to time-average estimates.

Q: what is the efficiency of the MWC and BEST traps? Refer to study that tests the efficiency of these instruments.

Q: can we assume the saltiphone is recording all particles impacting it? Is the momentum sufficient for all parts of the airborne grainsize curve to justify a linear relation between particle counts and flux?

Figure 6: clarify what BEST1, BEST2 are? (replicates?). What are box edges and error bars in box plots (e.g., percentiles?), perhaps define in caption. Replace ‘high’, and ‘low’ with shear velocities.

Figure 7: replace x, y, variables in the equations with the real variables used.

Figure 11: check equation in figure.

Table 2: normalize counts/s per area (e.g., divide by sampling area of saltiphone).

Line 267: would it be possible to split this section up with some sub-headings? For example the section talking about spatial variability could be a sub-heading.

Figure 14: what do the numbers on the boxplot represent?

Line 341 forward: Some photos or further detail about the site are needed here to make these data useful for other researchers. I kept on wishing for some context. What does this place look like? These data would be significantly more interesting to readers with some basic context.

Figure 15: perhaps here add a picture of the site.

Line 355: could you measure the fetch to help the readers understand whether this is playing a role?

Line 365: this sentence needs repair.

Lines 371-373: this could be quantified to make the point with much more impact.

---

## Round 0.2 · accepted · Accept

Thank you for the revising the manuscript